# Reply to Kawasaki et al. Comment on “Manole et al. Primary Pericardial Synovial Sarcoma: A Case Report and Literature Review. *Diagnostics* 2022, *12*, 158”

**DOI:** 10.3390/diagnostics14101013

**Published:** 2024-05-15

**Authors:** Simona Manole, Roxana Pintican, Emanuel Palade, Maria Magdalena Duma, Alexandra Dadarlat-Pop, Calin Schiau, Ioana Bene, Raluca Rancea, Diana Miclea, Viorel Manole, Adrian Molnar, Carolina Solomon

**Affiliations:** 1Department of Radiology, “Niculae Stancioiu” Heart Institute, 400001 Cluj-Napoca, Romania; simona.manole@gmail.com; 2Department of Radiology, “Iuliu Hatieganu” University of Medicine and Pharmacy, 400012 Cluj-Napoca, Romania; calin.schiau@yahoo.com (C.S.); ioanaboca90@yahoo.com (I.B.); 3Department of Cardiovascular and Thoracic Surgery, “Iuliu Hatieganu” University of Medicine and Pharmacy, 400012 Cluj-Napoca, Romania; paladeemanuel1@gmail.com (E.P.); adimolnar45@yahoo.com (A.M.); 4Department of Thoracic Surgery, Leon Daniello“ Pneumophtysiology Hospital Cluj-Napoca, 400332 Cluj-Napoca, Romania; 5Medimages Breast Center, 400458 Cluj-Napoca, Romania; magdaduma@gmail.com; 6Cardiology Department, Heart Institute “N. Stăncioiu”, 400001 Cluj-Napoca, Romania; dadarlat.alexandra@yahoo.ro (A.D.-P.);; 7Department of Internal Medicine, Faculty of Medicine, “Iuliu Hatieganu” University of Medicine and Pharmacy, 400012 Cluj-Napoca, Romania; 8Department of Medical Genetics, “Iuliu Hatieganu” University of Medicine and Pharmacy, 400012 Cluj-Napoca, Romania; bolca12diana@yahoo.com; 9Department of Cardiovascular Surgery, Heart Institute “N. Stăncioiu”, 400001 Cluj-Napoca, Romania; v_manole@yahoo.com

Thank you for your comment; it adds value to the article and highlights the importance of molecular testing. We agree with the authors. At that time, the only test that could be proposed in our lab was FISH testing to analyze translocation (X; 18). Translocation t(X;18) (p11.2; q11.2) is a chromosomal abnormality specific to synovial sarcoma, found only in synovial sarcoma and not in other types of tumors. It is present in most cases of synovial sarcoma (95%) [1]. The translocation involves a fusion between one of the SSX genes (SSX1, SSX2, or SSX4) and the SS18 gene (SYT), with the fusion protein participating in the formation of the SWI/SNF and Polycomb complexes, playing a role in epigenetic gene regulation.

However, there are a few cases (less than 5%) that do not exhibit this translocation, and other molecular modifications are also found. The other gene changes associated with synovial sarcoma include TP53, TERT, CDH1, CTNBB1, APC, HRAS, PTEN, PI3KCA, EGFR, BCL9, SETD2, TRRAP, PDGFRA, and NF1 [1,2]. Altered gene expression profiles have also been observed, such as in AURKA, KIF18A, Wnt (LEF1, TCF7, ZIC2, WNT5A, and FZD10), Hedgehog (PTCH1), NY-ESO-1 (CTAG1A), Notch (JAG1, JAG2, and HES1), and RTKs (FGF2, FGF3, EGFR, PDGFR, and IGFBP3). Altered gene expression sometimes leads to a more aggressive tumor with a higher potential for local recurrence and metastasis. Translocation (X; 18) is more common in younger patients with higher genomic stability, while other gene changes are more likely to be found in older patients with higher genomic instability.

The tumor morphology means that, in some cases, especially in monophase or poorly differentiated subtypes, molecular evaluation is necessary to establish the diagnosis as t(X; 18) has good sensitivity and specificity for diagnosing synovial sarcoma. In the described patient, the negative FISH test for this translocation did not allow for a clear differential diagnosis. Therefore, BCL-2 analysis was subsequently performed. FISH testing is comparable to RT-PCR for identifying the translocation [3]. Sequencing a panel of genes in the absence of the X; 18 translocation becomes useful for identifying the other gene changes potentially responsible for sarcoma development. Molecular diagnosis is also important for understanding the prognosis and choosing optimal and precise therapy.

The use of gene expression studies (RNAseq) is becoming useful for understanding the epigenetic mechanisms underlying synovial sarcoma and its prognosis and aggressiveness [4]. Therefore, we reiterate the significance of imaging in achieving an accurate diagnosis, particularly concerning the precise localization of pericardial tumors, including those of uncommon occurrence. It is essential to emphasize that molecular testing is equally important, providing a conclusive diagnosis that should not be marginalized.

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
