# Peer review of "Reply to Kawasaki et al. Comment on “Manole et al. Primary Pericardial Synovial Sarcoma: A Case Report and Literature Review. Diagnostics 2022, 12, 158”"

_diagnostics, 2024, doi:10.3390/diagnostics14101013_

Round 1

Reviewer 1 Report

Comments and Suggestions for Authors

Dear authors, 

Your reply (Reply to Tomonori Kawasaki et al. Comment on Manole et al. Primary Pericardial Synovial Sarcoma: A Case Report and Literture Review. Diagnostics 2022;12:158) provides an overview of the role of molecular testing in diagnosing synovial sarcoma, particularly the significance of the t(X;18)(p11.2;q11.2) translocation and other gene changes. 

While FISH testing is indeed valuable for identifying the t(X;18) translocation, it’s important to note that it may not detect all cases, as you’ve mentioned. This could potentially lead to false negatives, as in the case of the patient you described. 

I have three minor comments that should be addressed: 

1. While imaging is crucial for diagnosis, its role might be overstated in your reply. Molecular testing can provide a definitive diagnosis and should not be overshadowed. Kindly acknowledge these points in the concluding sentence of your comment.

2. Please include references in your comment. According to the MDPI guidelines and Diagnostics in Instructions for Authors section, References must be numbered in order of appearance in the text and listed individually at the end of the manuscript.

3. Minor revisions to the English language are necessary.

Kindly ensure that the aforementioned requests are included in the revised version.

Comments on the Quality of English Language

Minor revisions to the English language are necessary.

Author Response

Thank you very much for your comments:

  1. We rephrased it as follows: Thus, we reiterate the significance of imaging in achieving an accurate diagnosis, particularly concerning the precise localization of pericardial tumors, including those of uncommon occurrence. It is essential to underscore that molecular testing holds equal importance, providing a conclusive diagnosis that should not be marginalized
  2.  We added 4 references
  3. We revised the text and rephrased were was necessary.
